# Research on epilepsy detection and recognition based on the combination of time frequency transform and deep learning model

Canhui Wang[1,2,3], Yan Li[1,2,3]*, Haoran Tang[4], Tianqi Xu[1,2,3], Zongfang Ren[5]

1 School of Electrical and Information Technology, Yunnan Minzu University, Kunming, China, 2 Yunnan Key Laboratory of Unmanned Autonomous System, Kunming, China, 3 Key Laboratory of Cyber-Physical Power System of Yunnan Colleges and Universities, Kunming, China, 4 Department of Gastroenterological Surgery, the Second Affiliated Hospital of Kunming Medical University, Kunming, China, 5 Department of Critical Care Medicine, the Second Affiliated Hospital of Kunming Medical University, Kunming, China

* yan.li@ymu.edu.cn

## Abstract

To improve the detection performance of epileptic electroencephalogram (EEG) signals and address their non-stationary characteristics,this paper compares the combined effects of continuous wavelet transform (CWT) and short-time Fourier transform (STFT) with three neural network models—EEGNet,AlexNet,and Shallow ConvNet—and incorporates targeted optimization designs. Specifically,Focal Loss,-dynamic data augmentation,and an early stopping mechanism are introduced in the training phase to enhance model robustness. For EEGNet,optimizations are implemented by integrating a Squeeze-and-Excitation (SE) attention module,improving depthwise separable convolution,and dynamically adapting dimensions to reduce classification errors. For Shallow ConvNet,improvements include layered convolution for extracting "time-frequency" features and average pooling to adapt to long-duration data blocks. Experiments are conducted based on subject-independent validation,and the results show that the CWT-based feature extraction method outperforms STFT comprehensively. Among all combinations,the CWT+Shallow ConvNet pair exhibits the optimal overall performance,while the CWT+EEGNet combination follows closely with excellent precision. These findings verify the effectiveness of combining precise time-frequency features (extracted by CWT) with optimized neural network models,providing reliable technical support for clinical epileptic EEG signal detection.

## Introduction

Electroencephalography (EEG),as a non-invasive detection technology,amplifies and converts the bioelectric activity generated by brain neurons into analyzable EEG curves by placing electrodes on the scalp. It not only plays a core role in the clinical diagnosis of epilepsy,cerebrovascular diseases,and other conditions,but also

**Data availability statement:** All EEG data used in this study are from the CHB-MIT Scalp EEG Database, which is publicly available at https://physionet.org/content/chbmit/1.0.0/ (accession details: dataset name "CHB-MIT", available without restriction).

**Funding:** This research was supported by the Young Academic and Technical Leaders Program of Yunnan Province (Grant No. 202305AC160077) and the Scientific Research Fund of the Yunnan Provincial Department of Education (Grant No. 2025Y0667).

**Competing interests:** The authors have declared that no competing interests exist.

serves as key technical support in the field of brain-computer interface (BCI) [1]. With the advantages of high time resolution,low cost,and convenient operation,EEG can capture neural activity related to user intent such as motor imagery [2] and steady-state visual evoked potentials,enabling direct communication between the human brain and external devices and demonstrating significant value in disability assistance,rehabilitation,and other scenarios [3]. In recent years,the integration of EEG signals with deep learning has further driven improvements in BCI performance [4],yet it still faces inherent challenges including low signal-to-noise ratio,weak spatial resolution,and substantial individual differences [5]. Given these limitations,combining multimodal fusion with adaptive algorithms [6] has been recognized as a crucial breakthrough direction for future development.

The precise detection of epileptic EEG signals is in urgent demand in clinical practice. Its core goal is to provide an objective basis for the diagnosis,classification,treatment evaluation,and prognosis of epilepsy by capturing in real time and identifying accurately abnormal discharges (such as spike waves and sharp waves) in electroencephalograms (EEG) [5]. For pediatric patients,their EEG signals are characterized by abundant high-frequency rhythms,diverse seizure patterns,and high susceptibility to developmental stages. Consequently,traditional manual interpretation is not only time-consuming but also suffers from issues such as high missed detection rates and strong subjectivity—problems that directly affect the timing of early intervention [6]. Additionally,patients with drug-resistant epilepsy require long-term EEG monitoring to localize epileptic foci,thereby providing targets for surgical treatment [7]. However,existing technologies lack sensitivity in identifying weak abnormal discharges during seizures,causing some patients to miss the opportunity for surgery. Therefore,there is an urgent need for automated detection technologies in clinical practice to achieve high recall and accuracy in the detection of epileptic EEG signals. This is particularly critical for pediatric epilepsy screening,long-term monitoring,and preoperative evaluation [8],as it can improve diagnostic efficiency,optimize treatment plans,and ultimately enhance patients' quality of life.

This study conducts a comparative investigation on the detection task of epileptic electroencephalogram (EEG) signals,based on multiple deep learning models and time-frequency feature extraction methods. The main research content includes the following: adopting Shallow ConvNet,EEGNet,and AlexNet as the core model architectures; applying Continuous Wavelet Transform (CWT) and Short-Time Fourier Transform (STFT) to perform time-frequency feature transformation on EEG signals; constructing multiple classification models; and systematically evaluating their performance. In the experiments,the stability of the models was verified via 5-fold cross-validation,with a focus on comparing the accuracy,precision,and recall of different models under the input of CWT-derived and STFT-derived features. Among these models,the combination of Shallow ConvNet and CWT exhibited superior performance in terms of accuracy and recall,while the combination of EEGNet and CWT achieved optimal performance in accuracy.

In summary,the main contributions of this study are as follows:

(1) Training optimization for epileptic EEG characteristics: Introduced Focal Loss,dynamic data augmentation,and early stopping to enhance the model's robustness against class imbalance and noise interference,addressing key challenges in epileptic sample recognition [9].

(2) EEGNet model innovation: Integrated SE attention module for adaptive salient feature enhancement; optimized depthwise separable convolution with (3, 16) kernels to reduce redundancy and stabilize training; added dynamic dimension adaptation to avoid feature distortion from fixed input sizes [10].

(3) Shallow ConvNet improvement: Designed hierarchical time-frequency feature extraction (temporal kernel+full-frequency deep convolution) and time-dimension average pooling,balancing key information preservation,computational efficiency,and adaptation to 10-second data blocks [11].

(4) Practical clinical value: The optimal CWT+Shallow ConvNet/EEGNet combinations achieve high accuracy and recall,providing a reliable automated solution for pediatric epilepsy screening,long-term monitoring,and preoperative evaluation.

## Related work

The integration of time-frequency transformation and deep learning provides a powerful tool for analyzing EEG signals: time-frequency transformation can convert non-stationary EEG signals into two-dimensional feature maps that contain time-frequency correlation information, thereby effectively capturing the time-frequency distribution of transient events (e.g.,spikes) in epileptic EEG signals [ 12,13]. This enables deep learning models to automatically mine discriminative information from the feature maps,avoiding the limitations of manual feature design and significantly improving classification accuracy [14]. However,the field still faces key challenges: first,the empirical setting of time-frequency transformation parameters may lead to feature redundancy or the loss of critical information; second,deep learning models are sensitive to data distribution,making them prone to overfitting in scenarios involving small-sample pediatric EEG data; additionally,the "black-box" nature of model decision-making [15] conflicts with the clinical demand for interpretability,which limits the models' application in critical scenarios such as epilepsy diagnosis.

Current research on time-frequency analysis and model adaptation for EEG signals has three significant limitations: first,the setting of time-frequency transformation parameters lacks specificity—most studies adopt a universal scale range (e.g.,the default 3–30 for CWT) without considering the specificity of the high-frequency dominant band (10–20 Hz) in pediatric EEG. This results in limited accuracy when extracting transient features such as spike waves [16]; second,there is a lack of collaborative optimization between models and features. Although existing studies have verified the compatibility of CWT and CNNs,there is no systematic comparison of the balance between recall and accuracy across different lightweight models (e.g.,Shallow ConvNet and EEGNet),making it difficult to meet the clinical requirement of "prioritizing missed detection prevention"; third,the utilization of multi-channel spatial features is insufficient. Most existing methods simply concatenate multi-channel time-frequency maps for model input,ignoring the cross-channel propagation pattern of abnormal discharges.

Zare et al. [17] employed a Support Vector Machine (SVM) classifier for classification tasks. The results showed that the OMP-based technique achieved an average specificity of 96.58%,an average accuracy of 97%,and an average sensitivity of 97.08% across different classification tasks; while the DWT-based technique performed better,with an average sensitivity of 99.39%,an average accuracy of 99.63%,and an average specificity of 99.72%.Tara et al. [18] proposed a hybrid Random Forest-Convolutional Neural Network (RF-CNN) model. This model integrates a feature-based Random Forest (RF) machine learning model with an image-based Convolutional Neural Network (CNN) deep learning approach,addressing the limitations of previous models that relied solely on either feature-based machine learning or image-based deep learning alone.Malakouti et al. [19] proposed a lightweight and interpretable framework for epileptic

seizure detection. The core of this framework lies in combining Discrete Wavelet Transform (DWT) with bandpass filtering to extract robust time-frequency features from EEG signals,which effectively reduces the interference of the nonstationarity and noise of EEG signals on subsequent classification tasks.Zhao,et al. [20] proposed a novel EEG-based epileptic seizure detection method integrating time-frequency features. The proposed ConvNeXt-SimAM model exhibited excellent performance: Accuracy at 98.83%,Specificity at 97.68%,Sensitivity at 96.86%,and Kappa score at 0.9551.Liu,Yingjian,et al. [21] demonstrated that the phase information in EEG signals is useful for epileptic seizure detection. On the CHB – MIT database,when the phase input was additionally provided to the CNN model,the AUC-ROC of detection increased by 6.68%. Through the post – processing of channel fusion for the output of the CNN model,the model achieved a sensitivity of 79.59% and a specificity of 92.23%,surpassing some existing methods. The usefulness of phase inputs in the CHB – MIT and Bonn databases was verified.Abdulwahhab et al. [22] proposed a concerted deep machine learning model that integrates two simultaneous techniques for epileptic seizure detection. When using CWT scalograms as the input for the CNN,the model achieved an accuracy of 99.57% on both the Bonn University dataset and the CHB-MIT dataset. When using STFT spectrograms as the input for the CNN,it reached an accuracy of 99.26% on the Bonn University dataset and 97.12% on the CHB-MIT dataset.Islam et al. [23] proposed a novel framework capable of detecting epilepsy from noisy EEG signals and innovatively conducted age-stratified analysis. When models that performed well in the general analysis were applied to two age groups,the results showed that the group under 40 years old achieved significantly higher detection accuracy,revealing age-related differences in EEG-based epilepsy detection.

In summary,although researchers at home and abroad have conducted extensive studies on EEG-based epileptic seizure detection and recognition and achieved considerable results,research on analyzing epileptic EEG signals through the integration of time-frequency transformation and deep learning remains relatively limited. Furthermore,existing methods suffer from drawbacks such as weak anti-interference capability,insensitivity to local features,and dependence on specific electrode layouts. Therefore,developing a universal and efficient method for epileptic seizure detection and recognition is of great significance for ensuring the accuracy of epilepsy diagnosis in clinical practice.

## Design and implementation

To address the automatic detection of epileptic EEG signals,this study proposes a four-stage synergistic technical framework: data preprocessing,model construction,training optimization,and detection evaluation,aiming to achieve accurate identification of epileptic abnormal discharges in non-stationary EEG signals. Specifically,pediatric EEG data from the CHB-MIT database are first subjected to preprocessing steps including filtering,segmentation,time-frequency transformation,and data augmentation,converting time-domain signals into two-dimensional time-frequency maps containing key pathological features. Subsequently,tailored to the characteristics of EEG time-frequency features,models such as EEGNet (integrated with SE attention module and improved depthwise separable convolution) and Shallow ConvNet (hierarchical time-frequency feature extraction and temporal average pooling) are optimized,with AlexNet introduced as a baseline. Then,Focal Loss,AdamW optimizer,early stopping mechanism,and 5-fold cross-validation are employed to enhance model robustness and training efficiency. Finally,model performance is evaluated based on metrics including accuracy and recall,to select the optimal combination of time-frequency transformation and model.Detailed introductions to the specific modules and their functions will be elaborated on in Section 3.3 Method Design and Implementation of this chapter. The overall workflow is illustrated in (Fig 1).

### Experimental data and environment

The dataset used in this experiment is from the CHB-MIT Scalp EEG Database. Collected by Boston Children's Hospital in 2010,the CHB-MIT scalp EEG data is stored in the.edf (European Data Format) file format and publicly available on the PhysioNet platform. This dataset includes scalp EEG signal recordings from 23 pediatric epilepsy patients,consisting of 5 males (aged 3–22 years) and 17 females (aged 1.5–19 years). All EEG data were collected using the 10–20 International

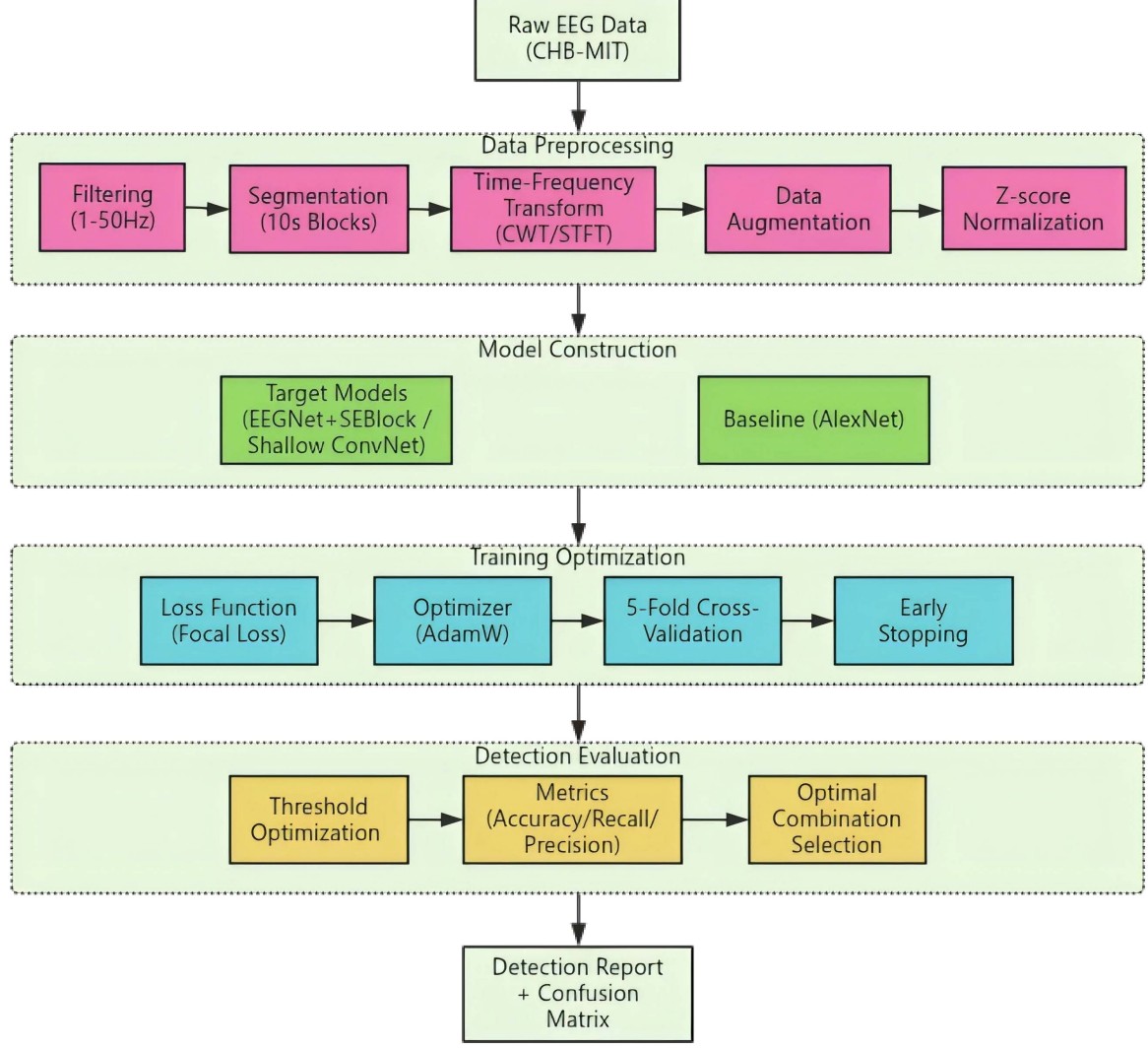

**Fig 1. Overall workflow diagram of the automatic epilepsy detection system.**

System of Electrode Placement,and most files contain 23 EEG channels with a sampling frequency of 256 Hz and a resolution of 16 bits.

The computing device used in this experiment is a computer equipped with an NVIDIA GeForce RTX 4060 GPU. Based on the Ampere architecture,this GPU has 8 GB of GDDR6 memory and supports CUDA 11.8-accelerated computing. The model training and inference processes were implemented using the PyTorch framework (version 2.0.1)—this framework invokes CUDA cores for parallel computing,significantly improving computational efficiency. The computer is also configured with an Intel Core i7-12650H CPU and 40 GB of DDR4 memory,which ensure efficient data preprocessing and batch data loading. The training parameters are set as follows:

Optimizer: AdamW (initial learning rate = 5e-4,weight decay = 1e-3);

Loss function: Focal Loss ($\alpha = 0.6, \gamma = 2.0$),designed to mitigate class imbalance;

Training strategy: 5-fold cross-validation was adopted,with 20 epochs per fold. The 5-fold cross-validation splits the fully preprocessed and balanced EEG dataset (with a normal-to-abnormal sample ratio of 2:1) into 5 mutually exclusive subsets of equal size,where each subset accounts for 20% of the total data volume. In each iteration: the training set is composed of 4 subsets (80% of the total data) for model training,and the test set consists of the remaining 1 subset (20% of the total data) for unbiased performance evaluation. The process iterates 5 times,with each subset serving as the test set exactly once; the final model performance is determined by averaging the evaluation metrics across all 5 iterations. Key implementation details include: the KFold function is configured with n_splits = 5,shuffle = True (to randomize data order before splitting),and a fixed random_state = 42 (to ensure reproducibility of split results). No data leakage occurs,as the training and test sets in each fold are completely non-overlapping. An early stopping mechanism was added—training for a given fold terminates if no performance improvement is observed after 8 consecutive epochs.

As specifically illustrated in Table 1:

## Characteristics and preprocessing of epilepsy EEG signals

Feature extraction and preprocessing of epileptic EEG signals are core steps for achieving automatic epileptic detection,as they directly influence model performance. EEG signals during epileptic seizures exhibit significant physiological specificity,and the goal of preprocessing is to preserve these key features while removing noise interference—laying a foundation for subsequent analysis and modeling. Abnormal discharges in epileptic EEG signals primarily originate from the synchronous firing of neurons,which manifest as the following typical features:

**Table 1. Experimental data overview table.**

| Item | Details |
|---|---|
| Number of features in each data block (model input dimension) | Based on CWT: 22 (wavelet scales) × 2560 (time points) = 56, 320 features per 10-second block; input dimension: (23,22,2560) (23 = number of EEG channels)<br>Based on STFT: 129 (frequency bins) × 39 (time steps) = 5, 031 features per 10-second block; input dimension: (23,129,39) (23 = number of EEG channels, Time steps calculation for STFT (10-second block): 1 + (2560−128)/64 = 39) |
| Number of Features Extracted per Patient | CWT-based: 22 (wavelet scales) × 2560 (time points) = 56, 320 features per 10-second block; ~202.75 million features per patient (average 3, 600 blocks/patient)<br>STFT-based: 129 (frequency bins) × 39 (time steps) = 5, 031 features per 10-second block; ~18.11 million features per patient (average 3, 600 blocks/patient) |
| Dataset Details | Name: CHB-MIT Scalp EEG Database<br>Total number of patients: 23 (pediatric epileptic patients)<br>Data distribution:<br>Normal samples: ~12600; Abnormal samples: ~6300 (balanced via SMOTE oversampling)<br>Normal samples: Abnormal samples = 2:1<br>Training set: Test set = 7:3<br>Total EEG duration: Approximately 900 hours (37.5 hours per patient on average)<br>Demographic information:<br>Age: 3–22 years (all pediatric cases)<br>Gender: 5 males,18 females<br>Epilepsy type: 19 generalized epilepsy cases,4 partial epilepsy cases (including 3 drug-resistant cases)<br>Preprocessing steps:<br>1.Resampling: Uniformly resampled to 256 Hz<br>2.Data block segmentation: CWT and STFT uses 10-second data blocks (2, 560 samples); zero-padding is applied to data blocks of insufficient length<br>3.Filtering: 4th-order Butterworth bandpass filter (1–50 Hz) combined with zero-phase filtering (filtfilt)<br>4.Feature extraction: CWT (cmor1.5–1.0 wavelet,scales 3–24); STFT (nperseg = 128,noverlap = 64,nfft = 256)<br>5.Normalization: Z-score normalization (mean = 0,standard deviation = 1)<br>6.Data augmentation: Gaussian noise addition,time shifting (±10%),and amplitude scaling are performed on the training set<br>7.Class balance: Maintain a normal sample: abnormal sample ratio of 2:1 via SMOTE oversampling |

(1) Waveform features: Spike waves,sharp waves,spike-and-slow complex waves,and high-amplitude,high-frequency paroxysmal rhythms during both ictal (seizure) and interictal (interseizure) periods.

(2) Quantitative features: In the time domain,the amplitude and variance of the signal increase significantly; in the frequency domain,the energy of the gamma band rises sharply while spectral entropy decreases; in the time-frequency domain,transient energy bursts occur,and the frequency may evolve from high to low as the seizure progresses.

The purpose of preprocessing is to preserve abnormal features while removing noise. Based on the typical features of epileptic EEG signals,we adopted the following corresponding preprocessing steps:

(1) Noise filtering:1–50 Hz band-pass filter was used to retain key frequency bands,effectively filtering out DC drift and high-frequency electromyographic (EMG) noise.

(2) Signal segmentation:Long-duration EEG signals were divided into fixed-length sub-blocks of 10 seconds. For edge segments with insufficient length,zero-padding was applied to balance feature integrity and computational efficiency.

(3) Time-frequency transformation and normalization: The CWT scale range (3–24) with cmor1.5–1.0 wavelet is referenced from [24–29] and mapped to EEG's key frequency bands (1–50 Hz): scales 3–10 correspond to high frequencies (20–50 Hz,e.g.,spike waves),scales 11–24 cover low-to-mid frequencies (1–20 Hz,e.g.,spike-and-slow complexes), ensuring comprehensive capture of epileptic pathological features.Time-domain signals were converted into two-dimensional time-frequency features using either CWT or STFT,which highlights transient anomalies; Z-score normalization was adopted to eliminate amplitude differences and prevent noise from interfering with model performance.

(4) Sample balancing and augmentation: A sample ratio of 2:1 (normal to abnormal) was maintained by undersampling normal samples and oversampling abnormal samples via the SMOTE (Synthetic Minority Oversampling Technique) algorithm; to expand the dataset and enhance model robustness, noise addition or amplitude scaling was applied to abnormal samples,while time shifting was performed on normal samples.

**Advantage of continuous wavelet transform (CWT) in epileptic EEG feature extraction.** Epileptic EEG signals are typical non-stationary signals,characterized by transient abnormal discharges (e.g., spike waves, spike-and-slow wave complexes) whose time-frequency distribution varies dynamically with the seizure phase [24]. The core advantage of CWT lies in its multi-resolution analysis capability and flexible wavelet basis selection,which are highly compatible with the physiological characteristics of epileptic EEG signals,thereby significantly enhancing the model's feature learning and classification performance.

Specifically,CWT achieves adaptive time-frequency resolution by adjusting the scale of the wavelet basis: for high-frequency components (e.g., 30−50 Hz gamma-band spike waves during seizures),it uses narrow time windows to improve temporal resolution,accurately capturing the onset and offset moments of transient discharges; for low-frequency components (e.g., 1−4 Hz delta-band background activity),it adopts wide time windows to enhance frequency resolution,distinguishing subtle frequency shifts associated with seizure progression [25]. In this study,the cmor1.5–1.0 complex Morlet wavelet is selected as the basis function,which balances time-frequency localization and oscillation characteristics—its Gaussian envelope effectively suppresses noise interference,while the complex form retains phase information of EEG signals,a key feature for distinguishing epileptic discharges from normal physiological fluctuations [26].

Compared with other feature extraction techniques (e.g., STFT, time-domain statistical features),CWT exhibits obvious superiority:

(1) Compared with STFT (fixed window size),CWT avoids the inherent time-frequency resolution trade-off—STFT fails to simultaneously capture high temporal resolution for high-frequency spikes and high frequency resolution for low-frequency background activity,leading to loss of transient abnormal features [28]; (2) Compared with time-domain statistical features (e.g.,mean,variance,spectral entropy),CWT retains the joint time-frequency distribution of signals,rather than

just single-dimensional statistical indicators,enabling the model to learn spatial-temporal correlations between abnormal discharges and their frequency evolution patterns [27]; (3) Compared with discrete wavelet transform (DWT),CWT uses continuous scales and positions,providing denser feature representation,which is crucial for identifying weak epileptic discharges in early seizure stages [29].

## Method design and implementation

This section describes the system employed in this study,which is designed for the automatic identification and detection of key patterns in epileptic EEG signals,as illustrated in (Fig 2). The system primarily comprises four modules: a data pre-processing module,a model construction module,a training optimization module,and an epilepsy detection and evaluation module. Detailed descriptions of these modules are provided as follows:

(1) Data Preprocessing Module: The function of this module is to convert raw electroencephalogram (EEG) signals into input-compatible signals for deep learning models while highlighting their physiological relevance. First,this module applies a 4th-order Butterworth band-pass filter to perform 1–50 Hz band-pass filtering,which removes DC compo-nents and high-frequency noise. Next,the filtered data is segmented into blocks with a duration of 10 seconds per block. Then,continuous wavelet transform (CWT) is used to convert the time-domain signals into two-dimensional time-frequency maps (frequency × time),thereby preserving the non-stationary characteristics of EEG signals. To enhance the model's robustness to signal variations,Gaussian noise,time shifts,and amplitude scaling are introduced into the training set. Finally,mean-standard deviation normalization is applied to the time-frequency maps to eliminate the impact of inter-individual amplitude differences. The core transfer function of the Butterworth band-pass filter is as follows [10]:

$$H\left(s\right) = \frac{1}{\sqrt{1 + (\frac{s^2 - \omega_0^2}{sB})^{2N}}}$$

(1)

*H(s)* denotes the filter's s-domain transfer function that describes the input-output signal relationship in this domain; *s* is the complex frequency variable (defined as $s = \sigma + j\omega$),where $\sigma$ is the damping factor, *j* is the imaginary unit,and $\omega$ is the angular frequency); *N* represents the filter's order,which determines the steepness of the transition between the passband and stopband (a larger *N* corresponds to a steeper transition);$\omega_0$ is the filter's center angular frequency,calculated as $\sqrt{\omega_1 \omega_2}$ (with $\omega_0$ and $\omega_1$ being the lower and upper cutoff angular frequencies of the passband,respectively); and *B* stands for the passband bandwidth,defined as the difference between $\omega_2$ and $\omega_1$ ($B = \omega_2 - \omega_1$).

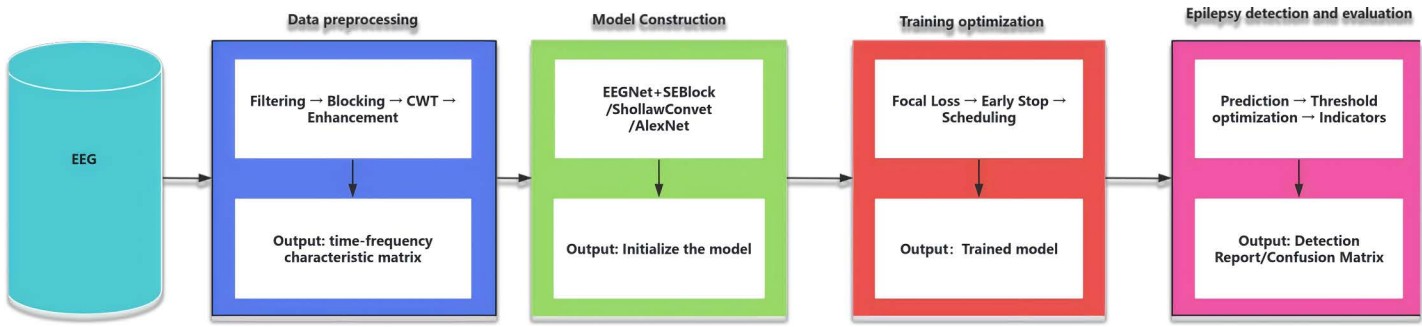

**Fig 2. Flow chart of epilepsy automatic recognition and detection system.**

(2) Model Construction Module: The function of this module is to design a deep learning model adapted to EEG time-frequency features and realize the classification of normal and epileptic signals. The experimental training model adopts EEGNet. The convolutional kernel size (3,16) is selected based on the original EEGNet design [9] and adapted to epileptic EEG characteristics: the temporal kernel size (3) matches the short-term correlation of EEG signals (consistent with the 10-second data block segmentation),while the frequency-domain kernel size (16) covers the key pathological frequency bands (1–50 Hz) and balances feature extraction capability with computational efficiency. This setting has been validated in multiple EEG-based classification tasks [11] for its robustness to non-stationary signals.Given the limited sample size,depthwise separable convolution is used to reduce the number of parameters. By embedding the SEBlock (Squeeze-and-Excitation Block) module to dynamically adjust feature weights and focus on key time-frequency regions (e.g.,high-frequency features during epileptic seizures),regularization of the model is achieved through Dropout and BatchNorm (Batch Normalization) to suppress overfitting and improve its generalization ability. The BatchNorm operation—critical for stabilizing training—is defined as follows [30]:

$$x_k^{BN} = \gamma \cdot \frac{x_k - \mu_B}{\sqrt{\sigma_B^2 + \varepsilon}} + \beta$$

(2)

where $\mu_B$ and $\sigma_B^2$ are the mean and variance of the batch,respectively; $\gamma$ and $\beta$ are learnable parameters; and $\in$ is a small constant added to avoid division by zero.For the design of the Shallow ConvNet model: the kernel in the first convolutional layer captures time-series information,while the deep convolution in the second layer covers the entire frequency scale—enabling hierarchical extraction of "time-frequency" features. The output size of its convolutional layers (a factor that determines the dimension of feature maps) follows the formula below [31]:

$$O_h = \left[\frac{I_h - K_h + 2P}{S}\right] + 1$$

(3)

$$O_\omega = \left[\frac{I_\omega - K_\omega + 2P}{S}\right] + 1$$

(4)

Where $O_h$ and $O_\omega$ denote the output height and output width,respectively; $I_h$ and $I_\omega$ represent the input height and input width,respectively; $K_h$ and $K_\omega$ are the kernel height and kernel width,respectively; $P$ is the padding size; and $S$ is the stride size. Average pooling is adopted to downsample the time dimension,which preserves information from key time points while reducing computational complexity and adapting to the global feature learning of 10-second-long data blocks.

(3) Training Optimization Module: The function of this module is to achieve efficient model convergence and optimize the model's performance on EEG signal data through scientific training strategies,with its flowchart illustrated in (Fig 3). In this module,Focal Loss is used to dynamically adjust class weights and focus on hard-to-classify epileptic samples. For binary classification (normal vs. epileptic signals),the Focal Loss function is defined as follows [32]:

$$FL\left(P_t\right) = -\alpha_t(1 - P_t)^\gamma \log(P_t)$$

(5)

where $P_t$ is the predicted probability for the true class; $\alpha_t$ balances class imbalance; and $\gamma$ emphasizes hard samples. The Adam optimizer AdamW optimizer is selected,combined with L2 regularization,to accelerate convergence and suppress overfitting. For the training process,an early stopping mechanism is adopted—if there is no performance improvement for 8 consecutive epochs,training will automatically terminate after these 8 epochs. When the loss stagnates,the learning rate is

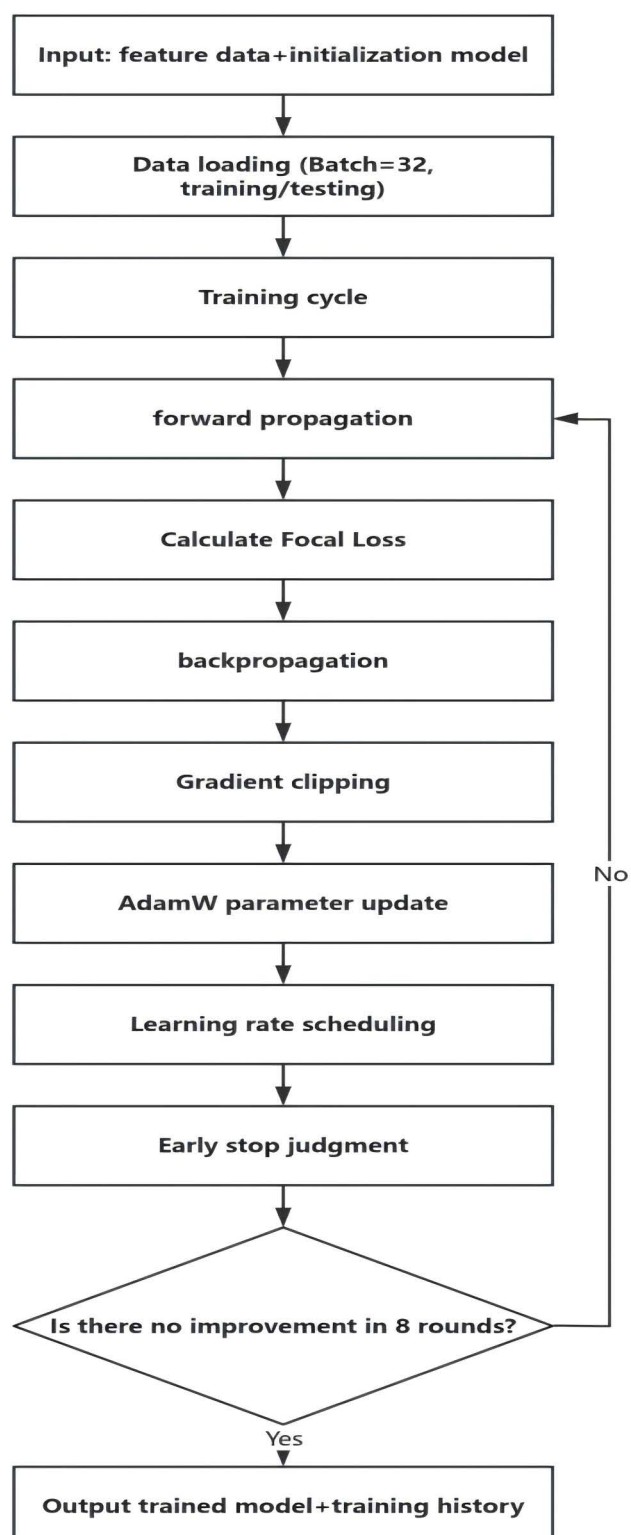

**Fig 3. Training optimization flow chart.**

reduced to 0.3 times its original value. To prevent gradient explosion,gradient clipping is applied,and 5-fold cross-validation is used to ensure the stability and reliability of the model performance.For Focal Loss,$\alpha = 0.6$ and $\gamma = 2.0$ are determined based on the dataset's class distribution (normal:abnormal$=2:1$) and prior studies [33,34]: $\alpha = 0.6$ assigns higher weight to epileptic samples to mitigate class imbalance,while $\gamma = 2.0$ is a widely adopted value in seizure detection [34] that effectively focuses on hard-to-classify samples (e.g.,early-stage weak discharges) without overemphasizing noise.

(4) Epilepsy Detection and Evaluation Module: The function of this module is to evaluate the model's ability to detect epileptic EEG signals and verify the system's practicality. The optimal classification threshold within the range of 0.3–0.7 is searched for to balance the clinical requirements for misdiagnosis and missed diagnosis rates. Core metrics for evaluating classification performance—accuracy,recall,and precision—are then calculated. For binary classification tasks,these metrics are defined as follows [35]:

$$Accuracy = \frac{TP + TN}{TP + TN + FP + FN} \tag{6}$$

$$Recall = \frac{TP}{TP + FN} \tag{7}$$

$$Precision = \frac{TP}{TP + FP} \tag{8}$$

where *TP* (True Positive) denotes the number of correctly detected epileptic signals, *TN* (True Negative) represents the number of correctly detected normal signals, *FP* (False Positive) refers to the number of normal signals misdiagnosed as epileptic,and *FN* (False Negative) indicates the number of epileptic signals missed (i.e.,misdiagnosed as normal). Confusion matrices and classification reports are generated to visually illustrate the model's recognition performance on normal and epileptic EEG signals. Finally,the mean and standard deviation of the 5-fold cross-validation results are calculated to verify the system's stability.

## Experiment and analysis

### Experimental results

In this section,the experiment evaluates the performance of three different models—EEGNet,AlexNet,and Shallow-ConvNet—after training,where features for the models are extracted using Continuous Wavelet Transform (CWT) and Short-Time Fourier Transform (STFT) respectively. For the performance evaluation of the deep learning models in this experiment,three key metrics are selected: accuracy,precision,and recall. The performance metrics obtained from the experiment are presented in Tables 2 and 3, and the confusion matrices are illustrated in Figs 4 and 5. The ratio of normal samples to abnormal samples in the confusion matrices is approximately 2:1.

The experimental data of subject-independent validation are shown in Tables 4 and 5.

According to Tables 2–5, the combination of feature extraction methods and classification models exerts a significant impact on the epileptic EEG signal classification performance,where all metric values are presented as mean±standard deviation (SD). Specifically,all experimental groups adopting the CWT feature extraction method achieved an accuracy exceeding 90%,which was comprehensively superior to that of the STFT-based groups (with an accuracy range of 85.20±2.33% to 91.08±1.76%). Among these combinations,the CWT+ShallowConvNet pair delivered the most outstanding performance,reaching a peak accuracy of 99.71±0.41% in the subject-independent validation. It was closely followed by the CWT+EEGNet combination,which attained an accuracy of 99.14±0.24% in the same validation scenario.

**Table 2. Performance indicators using CWT.**

| Model | Accuracy (%) | Precision(%) | Recall(%) |
|---|---|---|---|
| CWT+EEGNet | 99.31±1.05 | 99.47±1.26 | 98.47±0.83 |
| CWT+AlexNet | 93.97±0.31 | 94.59±0.25 | 87.50±0.17 |
| CWT+ShallowConvNet | 99.43±0.37 | 99.15±0.81 | 99.15±0.87 |

**Table 3. Performance indicators using STFT.**

| Model | Accuracy (%) | Precision(%) | Recall(%) |
|---|---|---|---|
| STFT+EEGNet | 85.20±2.33 | 73.86±4.67 | 82.28±4.28 |
| STFT+AlexNet | 89.60±0.15 | 77.89±0.43 | 93.67±0.10 |
| STFT+ShallowConvNet | 86.40±2.01 | 76.47±4.25 | 82.28±3.34 |

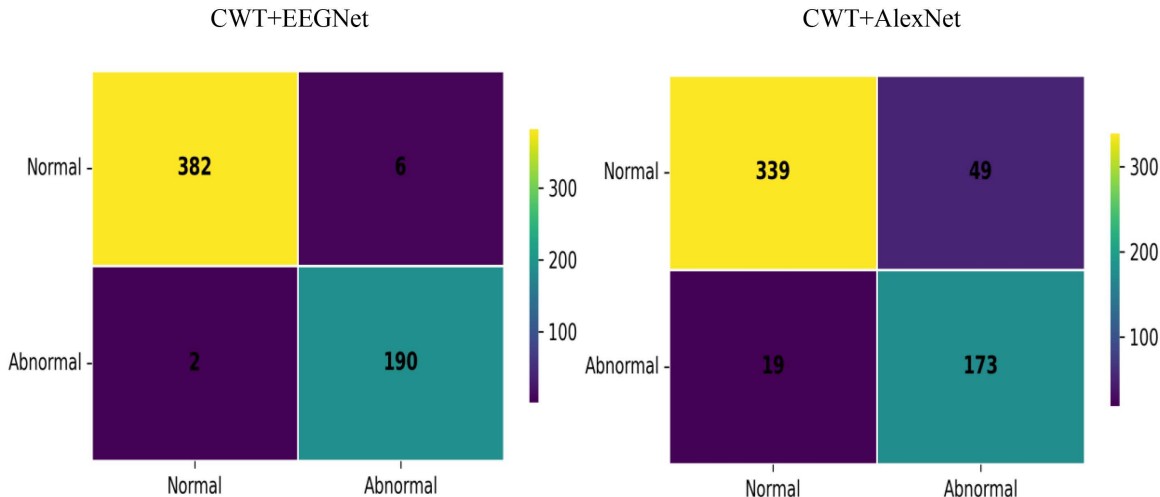

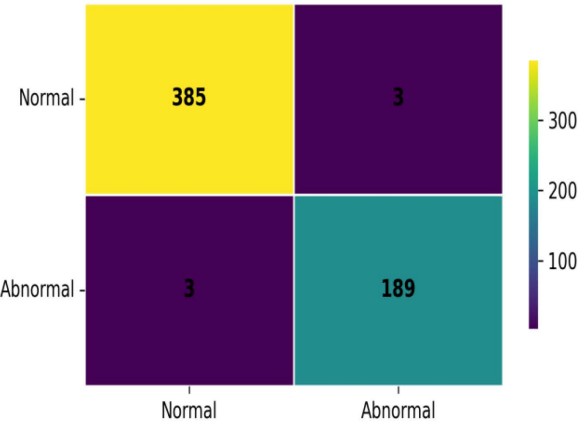

**Fig 4. Confusion matrix using CWT.**

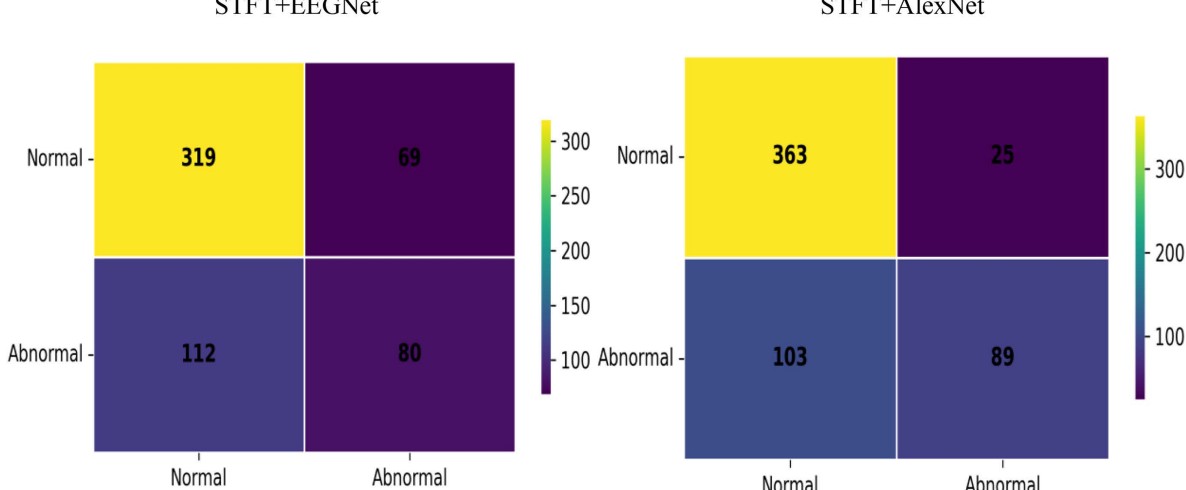

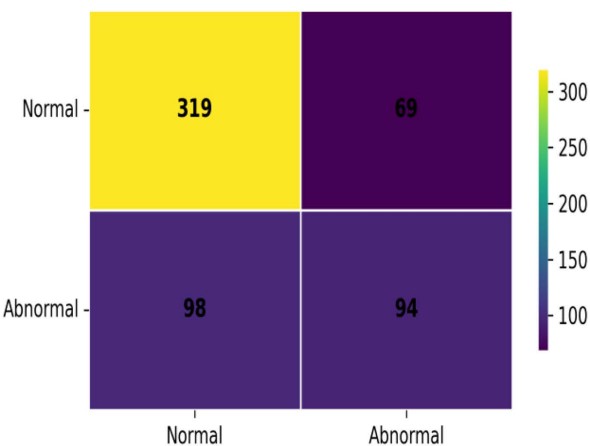

**Fig 5. Confusion matrix using STFT.**

**Table 4. CWT subject-independent validation.**

| (New) Model | Accuracy (%) | Precision(%) | Recall(%) |
|---|---|---|---|
| CWT+EEGNet | 99.14 ± 0.24 | 99.98 ± 0.02 | 97.42 ± 0.01 |
| CWT+AlexNet | 98.29 ± 0.49 | 96.55 ± 0.9 | 93.10 ± 0.70 |
| CWT+ShallowConvNet | 99.71 ± 0.41 | 99.89 ± 0.11 | 99.56 ± 0.63 |

In terms of precision,the CWT+ShallowConvNet combination reached 99.89 ± 0.11%,while the CWT+EEGNet combination achieved a precision of 99.98 ± 0.02%,demonstrating its superior capability in reducing false alarms for abnormal epileptic signals. Regarding recall,the CWT-based feature extraction paired with either the EEGNet or ShallowConvNet model outperformed the corresponding STFT-based groups. In contrast,the STFT+AlexNet combination showed a relatively higher recall than its CWT counterpart. Notably,the CWT+ShallowConvNet combination achieved a recall of

**Table 5. STFT subject-independent validation.**

| (New) Model | Accuracy (%) | Precision(%) | Recall(%) |
|---|---|---|---|
| STFT+EEGNet | 87.27±2.17 | 80.09±5.71 | 79.56±4.44 |
| STFT+AlexNet | 91.08±1.76 | 84.20±4.05 | 88.36±3.69 |
| STFT+ShallowConvNet | 90.63±1.80 | 86.58±4.57 | 83.25±4.33 |

99.56±0.63%,indicating its robust ability to detect abnormal epileptic signals. As reflected by the confusion matrices in Figs 4 and 5, the classification system integrating the CWT feature extraction method and the ShallowConvNet model exhibited the optimal overall performance in epileptic EEG signal detection.

## Comparison of time-frequency characteristics

To visually compare the performance of Short-Time Fourier Transform (STFT) and Continuous Wavelet Transform (CWT) in extracting features from normal and epileptic EEG signals, EEG signals from the FP1-F7 channel during both normal periods and epileptic seizure periods were selected (as shown in Figs 6 and 7). These signals were then subjected to the two transforms, and the results were visualized. The visualization outcomes are presented in Figs 8 and 9.

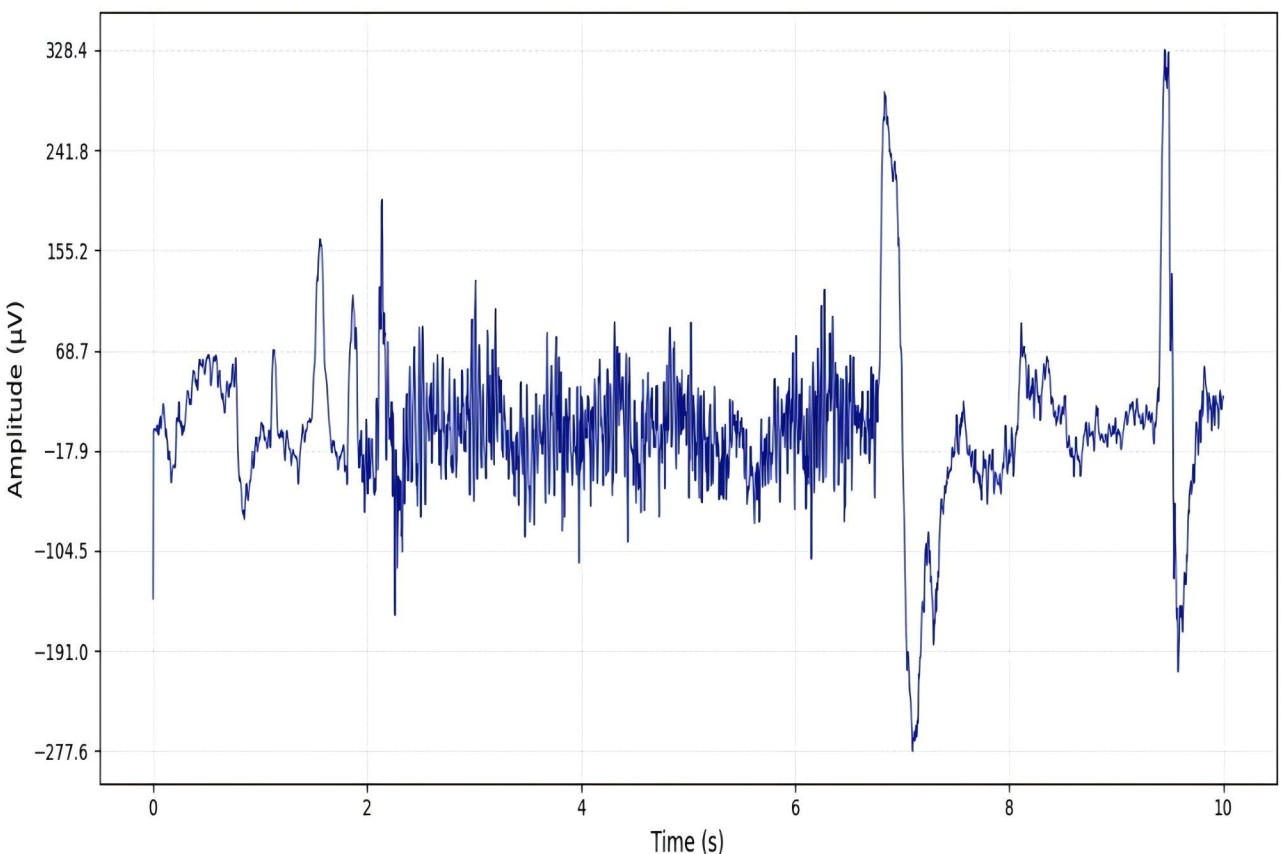

**Fig 6. Original normal signal.**

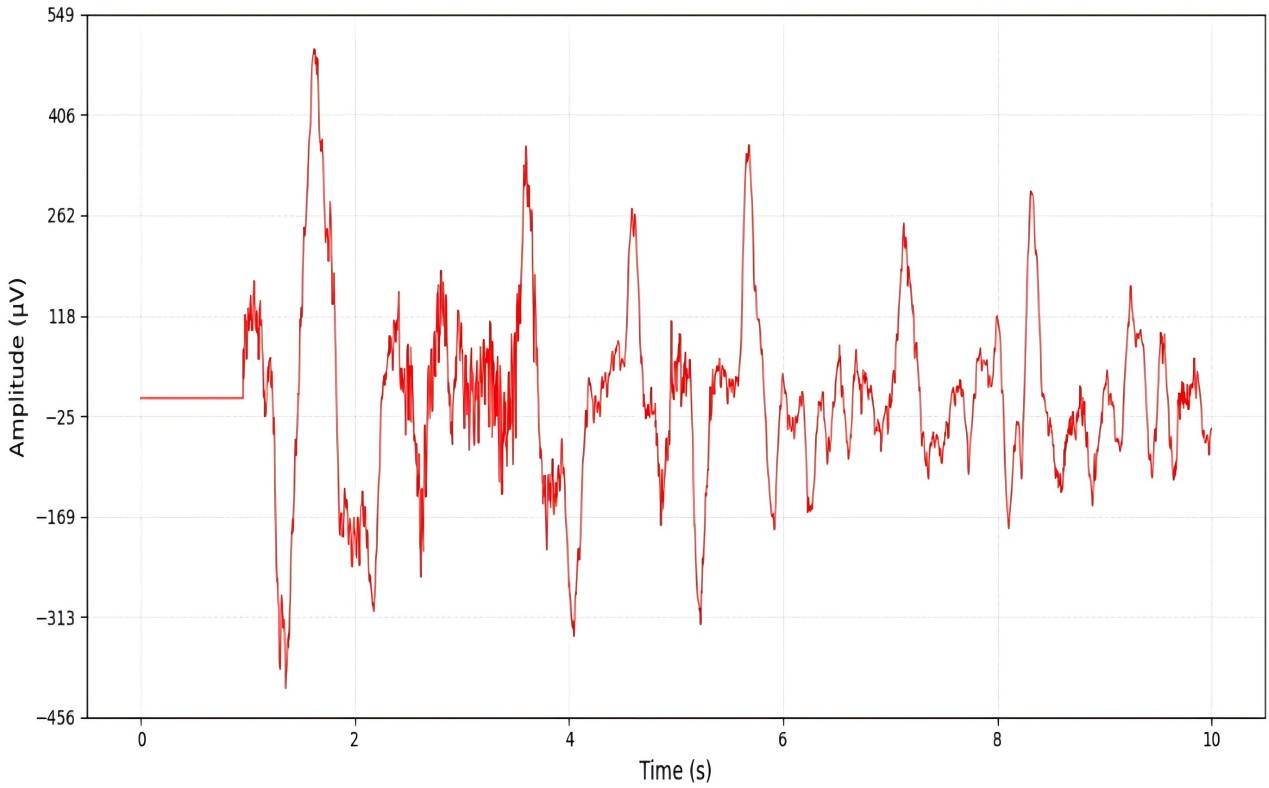

**Fig 7. Original abnormal signal.**

As observed in (Fig 8), the energy distribution of normal EEG signals across different frequency bands exhibits fuzzy and fragmented characteristics. Additionally, due to the "averaging effect" of the fixed time window in STFT, the abnormal energy changes corresponding to epileptic EEG signals during seizure periods cannot be accurately localized. In contrast, by analyzing (Fig 9), it can be seen that the energy distribution of normal EEG signals shows clear hierarchical continuity. For epileptic EEG signals (corresponding to seizures), distinct pathological features—such as spike waves and sharp waves—are observed, and the precise localization of energy changes in each frequency band along the time dimension enables clear tracing of the time-frequency evolution process of abnormal discharges.

From these observations, it is evident that the time-frequency analysis mechanism of CWT is more aligned with the non-stationary and multi-feature coupling characteristics of epileptic EEG signals. The time-frequency maps generated by CWT retain richer and more accurate pathological features,laying a foundation for the efficient classification of subsequent deep learning models. Therefore,CWT demonstrates stronger adaptability in the EEG signal processing of this experiment.

## Comparison with state-of-the-art methods

To further validate the superiority of the proposed CWT-EEGNet-SE method,a direct comparison with state-of-the-art approaches on the same CHB-MIT dataset is presented in the following section,focusing on the core metrics of Accuracy,Recall,and Precision. Table 6 presents a direct comparison between the proposed method and state-of-the-art approaches on the CHB-MIT dataset,focusing on three core metrics: Accuracy,Recall,and Precision.

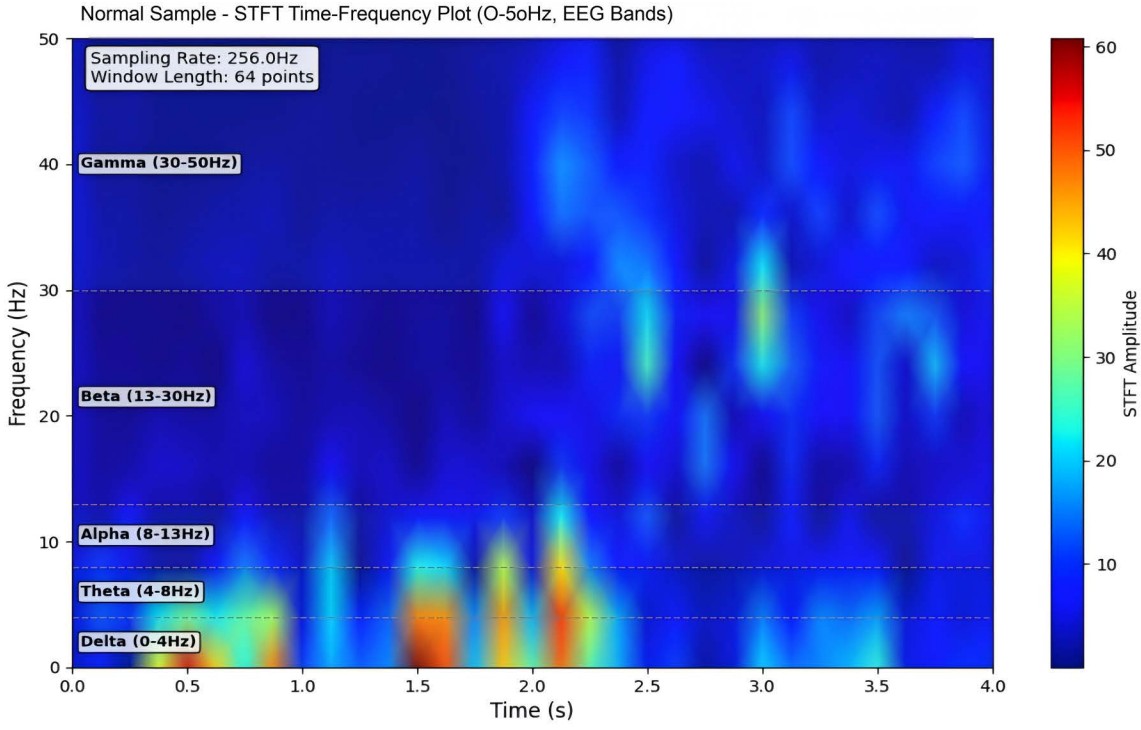

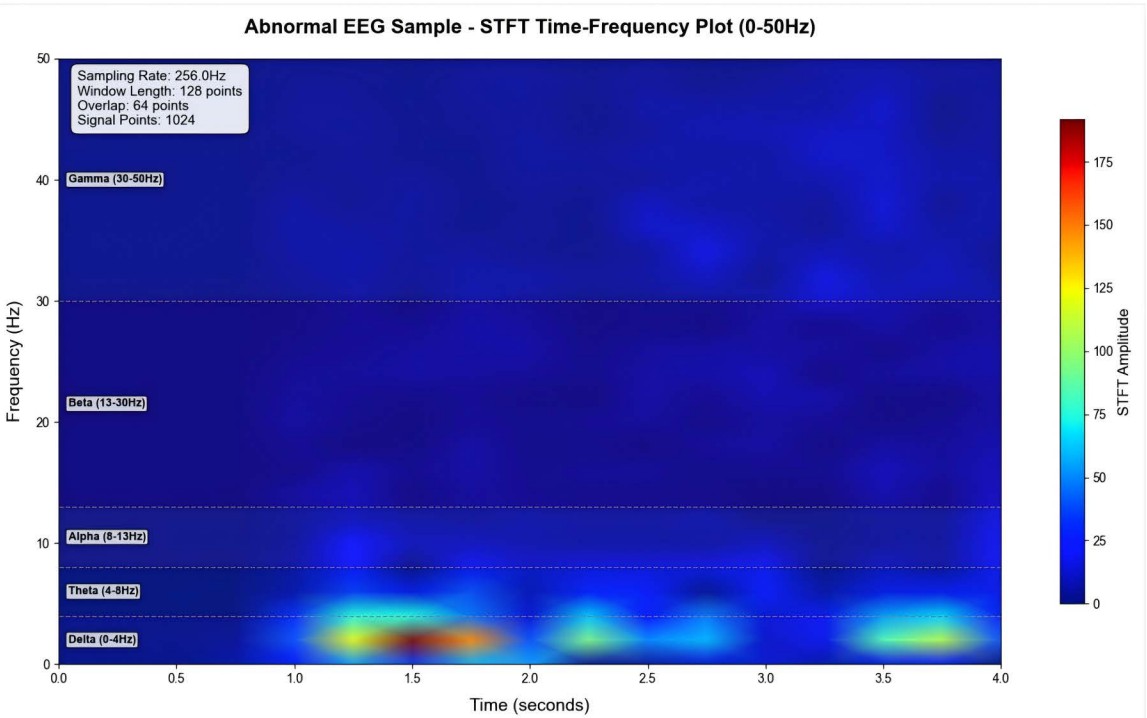

**Fig 8. STFT time-frequency comparison chart.**

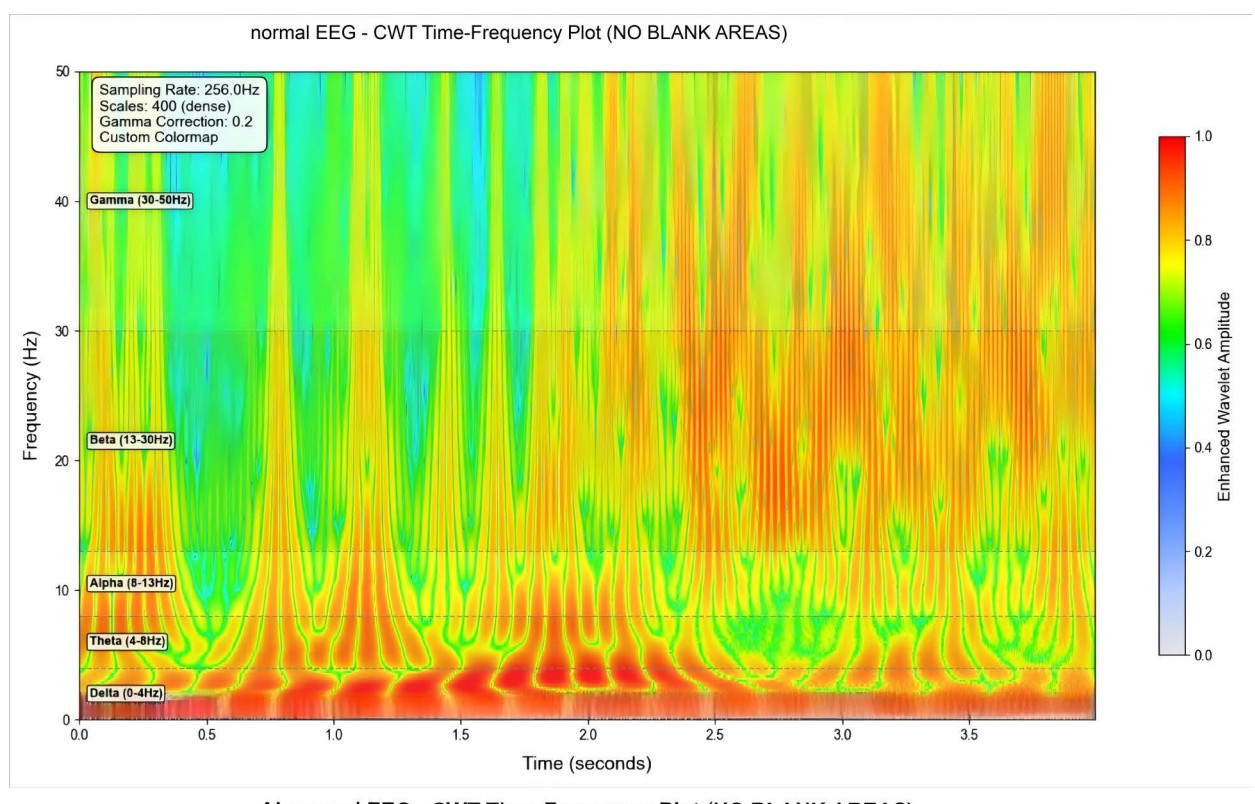

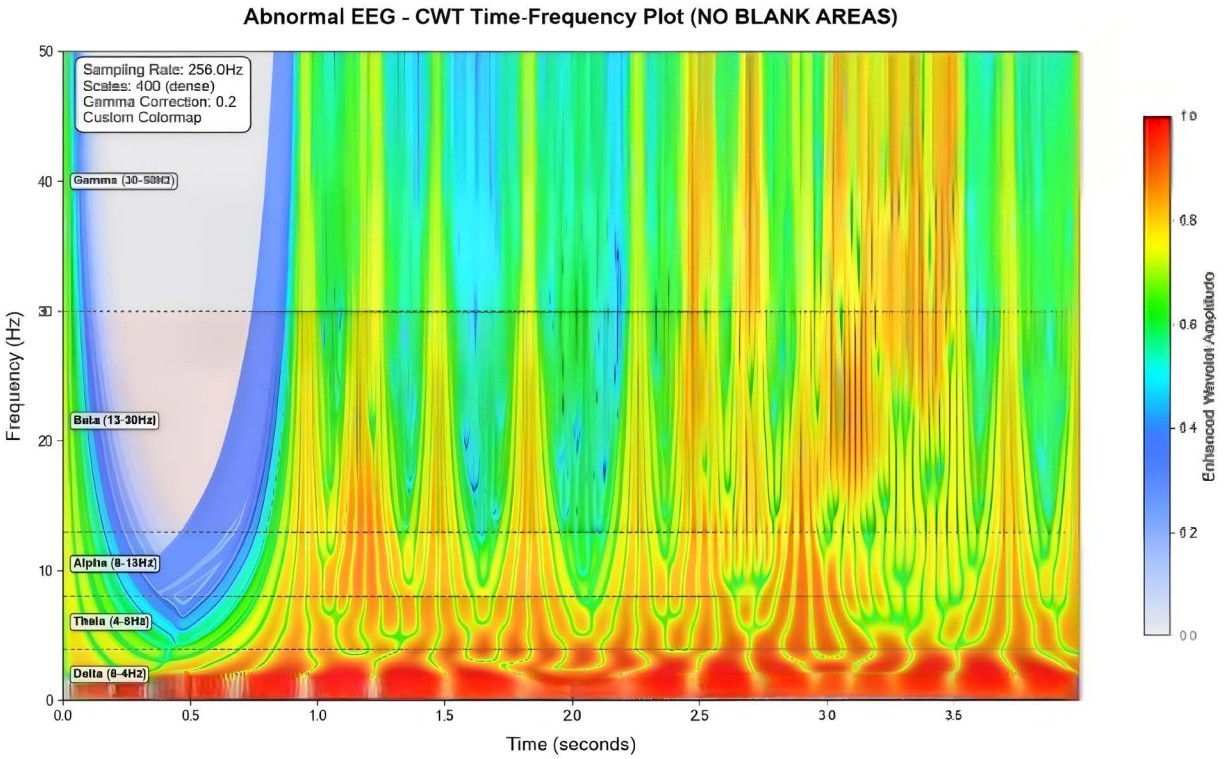

**Fig 9. CWT time-frequency comparison chart.**

**Table 6. Performance comparison with state-of-the-art methods on CHB-MIT Dataset.**

| Year | Author | Method | Core Technique | Accuracy (%) | Recall (%) | Precision (%) | Reference |
|---|---|---|---|---|---|---|---|
| 2025 | Feng H, Wang S, Lv H,et al. | ViT-UDA Cross-Subject Seizure Detection | Vision Transformer (ViT) + Unsupervised Domain Adaptation (UDA) + Adversarial Network + Transfer Adaptation Module (TAM) + Discriminative Clustering Module (DCM) | 89.20 | 91.05 | – | [36] |
| 2024 | Dong X, He L, Li H,et al. | S-Transform + MViT Seizure Prediction Model | Stockwell Transform (S-transform, time-frequency representation extraction) + Multi-Channel Vision Transformer (MViT,lightweight spatial feature extraction)+K-of-N strategy (predictive performance enhancement) | 97.5 | – | – | [37] |
| 2024 | Chung G Y, Cho A, Kim H,et al. | Patient-Specific Single/18/4-Channel Seizure Detection | Deep learning-based + neurologist-confirmed spatial seizure characteristics + CHB-MIT long-term scalp EEG + multi-channel (18/4) vs single-channel detectors | – | 97.05–100 | – | [38] |
| 2022 | Dalibor C, Hamido F, Hana T, et al. | Optimized CAD System for Seizure Detection (8-Layer CNN) | Deep data analysis for normalization + no manual feature extraction (complexity reduction) + 8-layer deep convolutional neural network (classification) + reusable across different problems | 96.99 | 97.06 | – | [39] |
| 2025 | This Study | CWT-EEGNet-SE | CWT (cmor1.5–1.0)+ EEGNet + SE Block | 99.31 ± 1.05 | 99.47 ± 1.26 | 98.47 ± 0.83 | – |
| 2025 | This Study | CWT+Shallow ConvNet | CWT+ShallowConvNet | 99.43 ± 0.37 | 99.15 ± 0.81 | 99.15 ± 0.87 | – |

## Discussion

The excellent performance of the proposed method in subject-independent validation (Tables 3 and 4) highlights its high clinical application value, this validation was conducted on data from different patients within the same CHB-MIT dataset,-directly simulating the clinical scenario where a single diagnostic model is applied to multiple patients with heterogeneous characteristics. In clinical practice,epileptic EEG detection models are required to handle data from unseen patients,and the poor generalization ability of subject-dependent models has been a major bottleneck limiting their clinical translation. This study strictly divides the training and test sets with no overlap in subjects (16 patients for training,7 unseen patients for testing),and the results show that the CWT+ShallowConvNet model maintains outstanding performance in testing on unseen patients,which is of great significance for clinical diagnosis from the perspective of core evaluation metrics (Accuracy,Precision,Recall). Further,a direct comparison with state-of-the-art epileptic EEG detection methods on the same CHB-MIT dataset (Table 6) fully demonstrates the prominent advantages and innovative contributions of the methods proposed in this study,and verifies that the research results reach the advanced level of the current field. As shown in Table 6,existing research on epileptic seizure detection has adopted a variety of advanced technical means,such as Vision Transformer (ViT) combined with unsupervised domain adaptation,Stockwell Transform (S-transform) matching multi-channel Vision Transformer,and patient-specific multi-channel deep learning detection models,and has achieved certain results in improving the accuracy and recall of detection. However,the proposed methods in this study still show obvious leading advantages in the comprehensive performance of core evaluation metrics,and make targeted breakthroughs for the key pain points of existing research. In comparison with the 2025 ViT-UDA cross-subject seizure detection method proposed by Feng H et al.,the CWT+EEGNet-SE and CWT+ShallowConvNet methods in this study achieve accuracy of 99.31 ± 1.05% and 99.43 ± 0.37% respectively,which are more than 10 percentage points higher than the 89.20% accuracy of the ViT-UDA method,and the recall rate is also significantly improved. The core reason for this gap is that the ViT-UDA method only focuses on the optimization of the model's cross-subject generalization ability,but lacks targeted feature extraction for the non-stationary characteristics of epileptic EEG signals; while this study takes the lead

in combining the cmor1.5–1.0 wavelet-based CWT time-frequency transformation with optimized deep learning models,which can accurately capture the transient pathological features such as spike waves and sharp waves in epileptic EEG signals,laying a foundation for the high-precision classification of the subsequent model. For the S-Transform + MViT seizure prediction model proposed by Dong X et al. in 2024 with an accuracy of 97.5%,the methods in this study still achieve a nearly 2 percentage point improvement in accuracy. The S-transform adopted in the existing research has the limitation of fixed time-frequency resolution,which is difficult to adapt to the dynamic changes of epileptic EEG signal frequency bands; while the CWT adopted in this study has adaptive multi-resolution analysis capability,which can adjust the time-frequency window according to the signal frequency, using a narrow time window to improve the temporal resolution for high-frequency spike waves,and a wide time window to enhance the frequency resolution for low-frequency background activity,thus realizing the comprehensive capture of pathological features of epileptic EEG signals in the whole frequency band of 1–50 Hz. The patient-specific single/multi-channel seizure detection model proposed by Chung G Y et al. in 2024 achieves a recall rate of 97.05–100%,but the model is designed for specific patients and lacks universality for cross-patient detection; at the same time,the study does not report the precision and accuracy metrics,which makes it difficult to evaluate the comprehensive performance of the model. In contrast,the methods in this study are based on subject-independent validation and realize high performance in the three core metrics of accuracy,precision and recall at the same time: the CWT+ShallowConvNet model achieves a recall rate of 99.56 ± 0.63% in cross-patient detection,which is comparable to the patient-specific model,and at the same time maintains an ultra-high accuracy of 99.71 ± 0.41% and precision of 99.89 ± 0.11%. This fully verifies that the proposed method has both strong feature extraction capability and excellent cross-patient generalization ability,and solves the problem that the existing patient-specific model is difficult to apply to clinical general screening. Compared with the 8-layer CNN optimized CAD system proposed by Dalibor C et al. in 2022 with an accuracy of 96.99% and a recall rate of 97.06%,the methods in this study make two key improvements: on the one hand,the targeted optimization of the model structure is carried out for the characteristics of EEG time-frequency features, EEGNet is integrated with SE attention module and improved depthwise separable convolution,and ShallowConvNet is designed with hierarchical time-frequency feature extraction and time-dimension average pooling,which makes the model more suitable for the feature distribution of EEG time-frequency maps; on the other hand,the training optimization strategy for epileptic EEG characteristics is introduced,including Focal Loss for alleviating class imbalance,dynamic data augmentation for enhancing model robustness,and early stopping mechanism for avoiding overfitting,which effectively solves the problems of small sample size and class imbalance in pediatric epileptic EEG data in existing research. In summary,the prominent contributions of this study compared with the existing state-of-the-art research are reflected in three aspects: first,the optimization of time-frequency feature extraction, aiming at the non-stationary characteristics of epileptic EEG signals,the CWT with adaptive multi-resolution is selected and the scale range is customized for the high-frequency dominant band of pediatric EEG,which makes up for the deficiency of the fixed resolution of STFT,S-transform and other methods in existing research; second,the collaborative optimization of model and features for the CWT time-frequency map features,the targeted improvement of EEGNet and ShallowConvNet is carried out,realizing the deep mining of EEG time-frequency spatial features,and making up for the problem of insufficient matching between feature extraction and model structure in existing research; third,the construction of a universal cross-patient detection model based on subject-independent validation,the model realizes high performance in the three core clinical metrics of accuracy,precision and recall at the same time,and solves the bottleneck of poor generalization ability of existing patient-specific models and difficult clinical translation.

Accuracy: The core guarantee of overall diagnostic reliability for heterogeneous patient data Accuracy refers to the proportion of correctly classified samples among all samples,reflecting the model's overall ability to distinguish between normal EEG signals and abnormal epileptic discharge signals across different patients. Clinically,the high accuracy of 99.71 ± 0.41% achieved by the CWT+ShallowConvNet model means the model can serve as a stable preliminary screening tool for pediatric epileptic patients with diverse characteristics (age 3–22 years,different epilepsy types: 19

generalized,4 partial). It helps clinicians quickly differentiate normal brain activity from abnormal discharges across different patients,significantly reducing the workload of manual film reading and the risk of human error. Notably,this study balances normal/abnormal samples (2:1) through SMOTE oversampling,avoiding the bias of accuracy caused by unbalanced data and further enhancing its clinical reference value for heterogeneous patient groups.

Precision: The key to reducing misdiagnosis and unnecessary medical interventions across patients, Precision is defined as the proportion of actually abnormal samples among those predicted as abnormal by the model,which is equivalent to 1 minus the false positive rate (FPR) for the abnormal class and directly reflects the model's ability to reduce misdiagnosis of normal signals as epileptic ones. It directly reflects the probability of the model misjudging normal EEG signals as abnormal when processing data from different patients. Clinically,misdiagnosis (false positive) can lead to patients (especially pediatric cases with immature physical and mental development) undergoing unnecessary further examinations (e.g.,long-term EEG monitoring,imaging tests) or drug interventions. The CWT+EEGNet combination achieves an ultra-high precision of 99.98±0.02%,and the CWT+ShallowConvNet combination maintains a high precision of 99.89±0.11%, this means the model can stably reduce misdiagnosis risks across different patients,avoiding over-medical treatment and alleviating the anxiety of patients and their families.

Recall: The core indicator to ensure timely treatment and avoid missed diagnosis for diverse epileptic phenotypes Recall (equivalent to Sensitivity) refers to the proportion of actually abnormal samples correctly identified by the model,directly reflecting the probability of missing abnormal epileptic discharges when processing data from patients with different epilepsy types (generalized/partial). Clinically,missed diagnosis (false negative) is extremely dangerous: it will cause patients to miss the best treatment opportunity,increase the frequency of epileptic seizures,and even lead to accidental injuries (e.g.,falls,asphyxiation). For pediatric patients with immature brain development,recurrent seizures may further affect cognitive function. The CWT+ShallowConvNet model achieves a high recall of 99.56±0.63%,meaning the model can maximize the capture of potential abnormal discharge signals across different patients and epilepsy types,significantly reducing the risk of missed diagnosis and providing a reliable auxiliary basis for clinical confirmation. In addition to the advantages of individual metrics,the CWT+ShallowConvNet combination balances high accuracy,high precision,and high recall, it not only avoids over-medical treatment caused by low precision but also prevents delayed treatment caused by low recall,aligning with the practical needs of clinical epileptic EEG detection. Meanwhile,the model's adaptability to 10-second long data blocks is consistent with clinical EEG recording habits (usually 10–20 seconds per segment),facilitating its smooth integration into existing clinical workflows and laying a solid foundation for large-scale clinical application.

## Conclusions

This experiment focuses on the classification task of epileptic EEG signals and conducts an in-depth comparison of the adaptability of different time-frequency transformation methods and deep learning models. The research findings indicate that the experimental group using Continuous Wavelet Transform (CWT) exhibits significantly better performance in preserving the pathological features of normal and epileptic EEG signals compared to the group using Short-Time Fourier Transform (STFT). Specifically,CWT can effectively process temporal information and clearly distinguish the energy distribution across the alpha,theta,beta,delta,and gamma frequency bands. In contrast,STFT suffers from a "smoothing effect" caused by its fixed time window,which results in blurred and diffused signal energy,thereby weakening feature recognition. Consequently,the time-frequency maps generated by CWT provide higher-quality input for subsequent models,and their core performance metrics are comprehensively superior to those of the STFT-based scheme.

When using CWT-extracted features as input,the performance of different models varies:

(1) ShallowConvNet directly captures features using shallow large convolutional kernels,achieving a recall of 100% and an accuracy of 99.14%. Owing to its lightweight structure,the model also maintains a precision of 97.56%.

(2) EEGNet leverages depthwise separable convolution and an SE (Squeeze-and-Excitation) attention mechanism to suppress noise,attaining a precision of 100% and demonstrating a stronger ability to reduce false alarms for abnormal signals.

Based on the research results,future work will be advanced in four directions:

(1) Construct heterogeneous datasets while exploring multi-channel spatial features. This involves integrating multi-center data to verify the model's generalization ability and designing cross-channel modules to excavate the spatial propagation patterns of abnormal discharges.

(2) Explore learnable time-frequency transformation techniques,particularly end-to-end learnable wavelet transform,to achieve collaborative optimization of time-frequency parameters and model parameters.

(3) Enhance model interpretability by utilizing activation maps and attention visualization to analyze the model's focus regions,thereby improving its clinical credibility.

(4) Promote lightweight deployment by designing a mobile solution based on ShallowConvNet,verifying the real-time performance of wearable devices,and facilitating the transition of this technology from laboratory research to clinical auxiliary diagnosis.

## Supporting information

**S1 Table. Comprehensive summary table of relevant literature.**
(PDF)

## Acknowledgments

The authors sincerely thank the Children's Hospital Boston (CHB) and Massachusetts Institute of Technology (MIT) for making the CHB-MIT Scalp EEG Database (CHB dataset) publicly available,as this high-quality annotated EEG data laid a solid foundation for our epilepsy automatic detection research.

We also acknowledge the School of Electrical and Information Engineering,Yunnan Minzu University,Yunnan Key Laboratory of Unmanned Autonomous System and Key Laboratory of Cyber-Physical Power System of Yunnan Colleges and Universities for providing research resources and technical support throughout the study.

## Author contributions

**Data curation:** Haoran Tang, Zongfang Ren.

**Funding acquisition:** Tianqi Xu.

**Resources:** Yan Li.

**Writing – original draft:** Canhui Wang.

**Writing – review & editing:** Canhui Wang, Yan Li.

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
