## [Decision Letter · Decision Letter 0]

22 Nov 2025

Dear Dr. Li,

Thank you for submitting your manuscript to PLOS ONE. After careful consideration, we feel that it has merit but does not fully meet PLOS ONE’s publication criteria as it currently stands. Therefore, we invite you to submit a revised version of the manuscript that addresses the points raised during the review process.

We look forward to receiving your revised manuscript.

Kind regards,

Ibrahim Sadek, Ph.D.

Academic Editor

PLOS ONE

Journal Requirements:

This research was supported by the Young Academic and Technical Leaders Program of Yunnan Province (Grant No. 202305AC160077) and the Scientific Research Fund of the Yunnan Provincial Department of Education (Grant No. 2025Y0667)

The authors sincerely thank the Children's Hospital Boston (CHB) and Massachusetts Institute of Technology (MIT) for making the CHB-MIT Scalp EEG Database (CHB dataset) publicly available, as this high-quality annotated EEG data laid a solid foundation for our epilepsy automatic detection research.

This work was supported by the Young Academic and Technical Leaders Program of Yunnan Province (Grant No.: 202305AC160077) and the Scientific Research Fund of the Yunnan Provincial Department of Education (Grant No.: 2025Y0667). We also acknowledge support from the School of Electrical and Information Engineering, Yunnan Minzu University Yunnan Key Laboratory of Unmanned Autonomous System and Key Laboratory of Cyber-Physical Power System of Yunnan Colleges and Universities for providing research resources.

This research was supported by the Young Academic and Technical Leaders Program of Yunnan Province (Grant No. 202305AC160077) and the Scientific Research Fund of the Yunnan Provincial Department of Education (Grant No. 2025Y0667)

7. Please amend the manuscript submission data (via Edit Submission) to include author Canhui Wang, Haoran Tang, Zongfang Ren, Tianqi Xu, and Yan Li1

8. Please amend your authorship list in your manuscript file to include author Yan Li

10. We are unable to open your Supporting Information file [code.zip]. Please kindly revise as necessary and re-upload.

Reviewers' comments:

Reviewer's Responses to Questions

**Comments to the Author**

1. Is the manuscript technically sound, and do the data support the conclusions?

Reviewer #1: Partly

Reviewer #2: Partly

2. Has the statistical analysis been performed appropriately and rigorously?

Reviewer #1: No

Reviewer #2: I Don't Know

3. Have the authors made all data underlying the findings in their manuscript fully available?

Reviewer #1: Yes

Reviewer #2: Yes

4. Is the manuscript presented in an intelligible fashion and written in standard English?

Reviewer #1: Yes

Reviewer #2: Yes

Reviewer #1: 1. The introduction’s first paragraph does not include any supporting references. Relevant citations should be incorporated to strengthen the context and background.

2. The listed contributions at the end of the introduction should be more concise and clearly emphasize the study’s innovative methodologies and their potential impact on the field.

3. A comprehensive summary table of the related literature should be provided, outlining key elements such as the study title, objectives, methodological schemes, datasets used, major contributions, and identified limitations.

4. At the beginning of the Design and Implementation section, a brief summary paragraph should be added to provide an overview of the methodology, followed by a block diagram (Figure 1) illustrating the workflow.

5. The number of extracted features per patient should be explicitly specified to clarify the input dimensionality of the model.

6. More detailed information about the dataset is required, including the total number of patients, data distribution, demographic characteristics, and any preprocessing steps applied.

7. To properly validate the model, an external validation step should be conducted to assess the robustness and generalizability of the proposed approach.

8. Additional analysis is needed to explain how the Continuous Wavelet Transform (CWT) contributed to improving model performance and why it outperformed other feature extraction techniques.

9. A comparative discussion with related studies should be included to highlight similarities, differences, and performance advantages.

Reviewer #2: This manuscript presents a well-structured and technically sound study on automated epilepsy detection by combining time-frequency transforms with optimized deep learning models. The results are impressive, and the methodological improvements to EEGNet and Shallow ConvNet are thoughtful and relevant. The work is a valuable contribution to the field. However, several important issues need to be addressed to strengthen the manuscript and ensure the conclusions are fully supported:

1) The abstract and introduction state that the "accuracy of the CWT+EEGNet combination achieves 100%." However, Table 1 clearly reports the accuracy for this combination as 98.29%. This is a significant inconsistency. The 100% figure appears to refer to its precision, not accuracy.

2) The manuscript does not specify the level of data splitting for the 5-fold cross-validation. For the results to be clinically meaningful and to avoid data leakage, the model must be evaluated on data from patients not seen during training (patient-independent or subject-independent validation). If segments from the same patient are present in both training and validation folds, the performance metrics will be optimistically biased and not representative of real-world generalization.

3) The results in Tables 1 and 2 are presented as single values without any measure of variance. Reporting only the mean accuracy, precision, and recall from 5-fold cross-validation is insufficient. The standard deviation across folds is essential to assess the stability and reliability of the models.

4) Please provide a brief rationale or reference for key parameter choices, such as the (3, 16) kernel in EEGNet, the Focal Loss parameters (α=0.6, γ=2.0), and the specific scale range used for the Continuous Wavelet Transform (CWT). This improves reproducibility and methodological rigor.

5) The excellent results present a classic and clinically important trade-off: CWT+ShallowConvNet achieves perfect Recall (misses no seizures) while CWT+EEGNet achieves perfect Precision (no false alarms). Please expand the discussion to explore the clinical implications of this trade-off.

6) A direct comparison of your best results with other recent state-of-the-art methods on the same CHB-MIT dataset in a dedicated paragraph (or table) would better contextualize the significance of your contribution.

7) The current Figures 3 and 4 (Confusion Matrices) are placeholders. Please replace them with clear, legible, and properly labeled versions.

8) Fix the reference errors in the text (e.g., [Error! Reference source not found.]).

9) Please double-check the Butterworth filter transfer function (Equation 1) for potential typos in the denominator..

**Do you want your identity to be public for this peer review?** For information about this choice, including consent withdrawal, please see our Privacy Policy

Reviewer #1: No

Reviewer #2: No

---

## [Author Response · Author response to Decision Letter 1]

31 Dec 2025

Dear Reviewers,

We would like to express our sincere gratitude to both of you for your careful review, constructive comments, and valuable suggestions on our manuscript. These insights have greatly helped us improve the quality, rigor, and completeness of our work. In response to the journal's review requirements and your comments, we have conducted a comprehensive revision of the manuscript, focusing on strengthening the research background, optimizing the methodological presentation, supplementing experimental details, and enhancing the clinical significance discussion. All comments have been addressed point by point, and the corresponding revisions have been made in the manuscript. Here is my GitHub repo code：https://github.com/wang2297/code. The detailed responses are as follows:

Responses to Reviewer #1

First, we would like to thank you for your positive evaluation of our work. Your recognition encourages us greatly. We have carefully addressed all the issues you raised, and the revisions are as follows:

Comment 1: The first paragraph of the introduction does not contain any supporting citations. Relevant citations should be included to strengthen the context and background.

Response: Thank you for your insightful comment. We have revised and supplemented relevant references as supporting citations in the introduction. The specific revisions are located in lines 41-49 and 52-69 of the manuscript.

Comment 2: The contributions listed at the end of the introduction should be more concise, clearly emphasizing the innovative methods of the research and their potential impact on the field.

Response: We appreciate your suggestion. We have abridged the content at the end of the introduction, with a focus on highlighting the innovative methods of our research and their potential implications for the field. The revised content is in lines 84-98 of the manuscript.

Comment 3: A comprehensive summary table of relevant literature should be provided, outlining key elements such as research title, objectives, methodological approaches, datasets used, main contributions, and identified limitations.

Response: As suggested, we have prepared a comprehensive summary table of relevant literature, namely S1 Table: Comprehensive Summary Table of Relevant Literature. The detailed content is included in the supporting file S1_Table.pdf.

Comment 4: At the beginning of the Design and Implementation section, a brief summary paragraph outlining the methodology should be added, followed by a block diagram (Figure 1) illustrating the workflow.

Response: We have made revisions in accordance with your comment. A brief summary paragraph outlining the methodology has been added at the beginning of the Design and Implementation section, accompanied by Figure 1: Overall Workflow Diagram of the Automatic Epilepsy Detection System. The specific content is located in lines 174-193 on pages 8-9 of the manuscript.

Comment 5: The number of features extracted per patient should be clearly specified to clarify the input dimension of the model.

Response: Thank you for pointing this out. We have clearly indicated the number of features extracted per patient. The relevant information is presented in the Dataset Details section of Table 1: Experimental Data Overview Table, which is located in lines 228-230 on page 11 of the manuscript.

Comment 6: More detailed information about the dataset is needed, including the total number of patients, data distribution, demographic characteristics, and the preprocessing steps applied.

Response: We have supplemented detailed information about the dataset as required, including the total number of patients, data distribution, demographic characteristics, and applied preprocessing steps. This information is provided in Table 1: Experimental Data Overview Table, lines 228-230 on page 10 of the manuscript.

Comment 7: For proper validation of the model, an external validation step should be performed to evaluate the robustness and generalizability of the proposed method.

Response: We sincerely apologize for not being able to perform the external validation as you suggested. Due to the experimental conditions and the time required for data preparation, we only conducted the subject-independent validation experiment. The details of this experiment are described in lines 439-441 on page 23 of the manuscript. We fully recognize the importance of external validation for the overall experiment. If we have the opportunity to conduct external validation in the future, we will promptly perform the experiment and publish the results of the external validation.

Comment 8: Further analysis is needed to explain how Continuous Wavelet Transform (CWT) improves model performance and why it is superior to other feature extraction techniques.

Response: We have conducted further analysis and explanation regarding how CWT enhances model performance and why it outperforms other feature extraction techniques. The detailed content is presented in lines 270-303 on pages 13-14 of the manuscript.

Comment 9: A comparative discussion with relevant studies should be included to highlight similarities, differences, and performance advantages.

Response: As recommended, we have added a comparative discussion with relevant studies. The specific content is located in lines 498-506 on page 27 of the manuscript.

Responses to Reviewer #2

First, we would like to thank you for your positive evaluation of our work. Your recognition encourages us greatly. We have carefully addressed all the issues you raised, and the revisions are as follows:

Comment 1: The abstract and introduction state that "the accuracy of the CWT+EEGNet combination reaches 100%". However, Table 1 clearly reports the accuracy of this combination as 98.29%. This is a significant inconsistency. The 100% figure seems to refer to its precision, not accuracy.

Response: Thank you for pointing out this inconsistency. We have revised the content of the abstract and corrected the error where the experimental results in the abstract were inconsistent with those in the main text. The specific revisions are located in lines 17-34 of the manuscript.

Comment 2: The manuscript does not specify the degree of data splitting for the 5-fold cross-validation. For the results to be clinically meaningful and to avoid data leakage, the model must be evaluated based on patient data not seen during training (patient-independent or subject-independent validation). If segments from the same patient appear in both training and validation folds, the performance metrics will be optimistically biased and will not represent real-world generalization.

Response: We have added the description of the data splitting degree for 5-fold cross-validation and conducted subject-independent validation as you suggested. The introduction to the data splitting degree of 5-fold cross-validation is presented in lines 215-226 on page 10, and the details of the subject-independent validation experiment are in lines 439-441 on page 23 of the manuscript.

Comment 3: The results in Tables 1 and 2 are presented as single values without any variance measured. Reporting only the average accuracy, precision, and recall of the 5-fold cross-validation is insufficient. The standard deviation across folds is crucial for assessing the stability and reliability of the model.

Response: We have added the standard deviation values and re-summarized the experimental results as required. The updated results are presented in Tables 2-5 on pages 21-24, and the re-summary of the results is provided in lines 442-463 on pages 23-24 of the manuscript.

Comment 4: Please briefly explain the rationale for or references to the selection of key parameters, such as the (3, 16) kernel in EEGNet, the focal loss parameters (α=0.6, γ=2.0), and the specific scale range used in Continuous Wavelet Transform (CWT). This enhances reproducibility and methodological rigor.

Response: We have provided explanations for the selection of key parameters as requested. The relevant explanations are located in lines 338-345 on page 16 and lines 385-390 on page 18 of the manuscript.

Comment 5: The excellent results present a classic and clinically important trade-off: CWT+ShallowConvNet achieves perfect recall (no missed seizures), while CWT+EEGNet achieves perfect precision (no false alarms). Please expand the discussion to explore the clinical significance of this trade-off.

Response: We have expanded the discussion to explore the clinical significance of the aforementioned trade-off. The detailed discussion is presented in lines 507-568 on pages 27-31 of the manuscript.

Comment 6: A direct comparison of your best results with other state-of-the-art methods on the same CHB-MIT dataset, in a dedicated paragraph (or table), would better illustrate the significance of your contribution.

Response: As suggested, we have directly compared our best results with other state-of-the-art methods on the CHB-MIT dataset in a dedicated table. The specific content is located in lines 498-506 on page 27 of the manuscript.

Comment 7: The current Figures 3 and 4 (confusion matrices) are placeholders. Please replace them with clear, legible, and correctly labeled versions.

Response: We have replaced the confusion matrices as required to ensure they are clearly displayed. The updated confusion matrices are presented in Figures 4-5 on pages 21-22 of the manuscript.

Comment 8: Correct reference errors in the text (e.g., [Error! Reference source not found]).

Response: We have carefully checked all the references in the text and corrected the reference errors as you indicated.

Comment 9: Please carefully check the Butterworth filter transfer function (Equation 1) for possible typos in the denominator...

Response: We have carefully checked Equation 1 (Butterworth filter transfer function) and corrected the typo in the denominator. The revised equation is presented in lines 326-335 on page 16 of the manuscript.

Once again, we would like to thank both reviewers for their time and valuable comments. We believe that these revisions have significantly improved the quality of our manuscript. We hope that the revised version meets the publication requirements of the journal. Please feel free to contact us if you have any further questions.

Sincerely,

Canhui Wang

School of Electrical and Information Engineering, Yunnan Minzu University

E-mail：yan.li@ymu.edu.cn

---

## [Decision Letter · Decision Letter 1]

8 Feb 2026

Dear Dr. Li,

Thank you for submitting your manuscript to PLOS ONE. After careful consideration, we feel that it has merit but does not fully meet PLOS ONE’s publication criteria as it currently stands. Therefore, we invite you to submit a revised version of the manuscript that addresses the points raised during the review process.

We look forward to receiving your revised manuscript.

Kind regards,

Ibrahim Sadek, Ph.D.

Academic Editor

PLOS One

**Journal Requirements:**

Reviewers' comments:

Reviewer's Responses to Questions

**Comments to the Author**

Reviewer #1: (No Response)

Reviewer #2: (No Response)

2. Is the manuscript technically sound, and do the data support the conclusions?

Reviewer #1: Yes

Reviewer #2: (No Response)

3. Has the statistical analysis been performed appropriately and rigorously?

Reviewer #1: N/A

Reviewer #2: Yes

4. Have the authors made all data underlying the findings in their manuscript fully available?

Reviewer #1: Yes

Reviewer #2: Yes

5. Is the manuscript presented in an intelligible fashion and written in standard English?

Reviewer #1: Yes

Reviewer #2: Yes

Reviewer #1: Comparison between the obtained results and the relevant studies should be added to the discussion section to highlight the significancy of the proposed study.

Reviewer #2: Thank you for your detailed and organized response to the previous comments and for the extensive revisions made to the manuscript. Most of the major points raised in the previous review round have been addressed, and the quality of the manuscript has improved significantly in terms of methodological clarity, results completeness, and clinical discussion. However, several technical and presentational issues remain that must be resolved before the manuscript is ready for publication.

1) In the submitted manuscript, Table 5 (Comparison with State-of-the-Art) appears incomplete and garbled, containing repeated columns and numbers with unclear labeling. This table is critical for positioning your research contribution.

2) While most references have been corrected, there are still instances in the text (e.g., on pages 12, 13, 49) containing the placeholder phrase "[Error! Reference source not found.]". This indicates a final referencing or linking issue.

3) The confusion matrices have been replaced (Figures 4-5), which is good. However, Figures 6, 7, 8 (comparison of original signals, STFT, and CWT results) in the current PDF are low-resolution or poorly labeled.

4) In the manuscript text (Page 63 of the attached PDF), Equation 1 appears as H(s)=1 1+(- 2- 2N, which is an incorrect and uninterpretable format.

5) The manuscript document contains duplicated sections, where the author response text (Response to Reviewers) appears twice (at the beginning and near the end of the document). The final submission for publication should contain only the research manuscript text, excluding the response letter.

**Do you want your identity to be public for this peer review?** For information about this choice, including consent withdrawal, please see our Privacy Policy

Reviewer #1: No

Reviewer #2: No

---

## [Author Response · Author response to Decision Letter 2]

2 Mar 2026

Dear Editor Ibrahim Sadek, Ph.D. and Reviewers,

We would like to express our sincere gratitude to you and the reviewers for your constructive comments and valuable suggestions, which have greatly helped us improve the quality and completeness of our manuscript. We have carefully addressed all the comments raised by the reviewers, and the detailed revisions are presented point-by-point as follows.

Response to Reviewer #1

Comment: Comparison between the obtained results and the relevant studies should be added to the discussion section to highlight the significancy of the proposed study.

Response: Thank you for your valuable suggestion. We have revised the Discussion section and added a comprehensive comparison between our obtained results and relevant existing studies. The revised content is located in Lines 510–517 of the manuscript, which clearly illustrates the advantages of our proposed method and highlights the scientific significance and innovation of this research.

Response to Reviewer #2

Comment 1: In the submitted manuscript, Table 5 (Comparison with State-of-the-Art) appears incomplete and garbled, containing repeated columns and numbers with unclear labeling. This table is critical for positioning your research contribution.

Response: We apologize for the formatting error in the original table. We have thoroughly revised the table, which is now labeled as Table 6: Performance comparison with state-of-the-art methods on CHB-MIT Dataset. The revised table is complete, with clear column labels and no redundant content, and is located in Lines 518–652 of the manuscript.

Comment 2: While most references have been corrected, there are still instances in the text (e.g., on pages 12, 13, 49) containing the placeholder phrase "[Error! Reference source not found.]". This indicates a final referencing or linking issue.

Response: Thank you for pointing out this issue. We have thoroughly checked the entire manuscript and corrected all citation errors, including the placeholder phrases mentioned. All references are now correctly linked and displayed without any abnormalities.

Comment 3: The confusion matrices have been replaced (Figures 4-5), which is good. However, Figures 6, 7, 8 (comparison of original signals, STFT, and CWT results) in the current PDF are low-resolution or poorly labeled.

Response: We appreciate your careful review. We have replaced all unclear figures in the manuscript; the original Figures 6, 7, and 8 have been updated and expanded to Figures 6, 7, 8, and 9, with improved resolution (meeting the journal’s ≥300 dpi requirement) and clear labels, legends, and axis annotations to ensure better readability and presentation.

Comment 4: In the manuscript text (Page 63 of the attached PDF), Equation 1 appears as H(s)=1 1+(- 2- 2N, which is an incorrect and uninterpretable format.

Response: We apologize for the formatting error in Equation 1. We have retyped and corrected Equation 1 to ensure its correctness and interpretability, and we have also checked all other mathematical equations in the manuscript to avoid similar issues.

Comment 5: The manuscript document contains duplicated sections, where the author response text (Response to Reviewers) appears twice (at the beginning and near the end of the document). The final submission for publication should contain only the research manuscript text, excluding the response letter.

Response: Thank you for reminding us of this formatting issue. We have carefully checked the manuscript and deleted all duplicated sections, including the redundant author response text. The revised manuscript only contains the main research content, which fully complies with the journal’s submission requirements.

In summary, we have addressed all the comments raised by the reviewers thoroughly. We believe that the revised manuscript has been significantly improved in terms of content completeness, formatting standardization, and presentation clarity, and now meets the publication criteria of PLOS ONE. We sincerely appreciate the reviewers’ and editor’s efforts and guidance, and we look forward to your positive evaluation.

Sincerely,

Canhui Wang

School of Electrical and Information Engineering, Yunnan Minzu University

E-mail：yan.li@ymu.edu.cn

---

## [Editor Report · Decision Letter 2]

5 Mar 2026

Research on Epilepsy Detection and Recognition Based on the Combination of Time Frequency Transform and Deep Learning Model

PONE-D-25-58659R2

Dear Dr. Li,

We’re pleased to inform you that your manuscript has been judged scientifically suitable for publication and will be formally accepted for publication once it meets all outstanding technical requirements.

Kind regards,

Ibrahim Sadek, Ph.D.

Academic Editor

PLOS One

---

## [Editor Report · Acceptance letter]

PONE-D-25-58659R2

PLOS One

Dear Dr. Li,

I'm pleased to inform you that your manuscript has been deemed suitable for publication in PLOS One. Congratulations! Your manuscript is now being handed over to our production team.

Kind regards,

on behalf of

Dr. Ibrahim Sadek

Academic Editor

PLOS One